

# MeteoMex: open infrastructure for networked environmental monitoring and agriculture 4.0

Robert Winkler[1,2]

[1] Kuturabi S.A. de C.V., Irapuato, Mexico
[2] Department of Biotechnology and Biochemistry, CINVESTAV Unidad Irapuato, Irapuato, Mexico

## ABSTRACT

Air, water, and soil are essential for terrestrial life, but pollution, overexploitation, and climate change jeopardize the availability of these primary resources. Thus, assuring human health and food production requires efficient strategies and technologies for environmental protection. Knowing key parameters such as soil moisture, air, and water quality is essential for smart farming and urban development. The MeteoMex project aims to build simple hardware kits and their integration into current Internet-of-Things (IoT) platforms. This paper shows the use of low-end Wemos D1 mini boards to connect environmental sensors to the open-source platform ThingsBoard. Two printed circuit boards (PCB) were designed for mounting components. Analog, digital and $I^2C$ sensors are supported. The Wemos ESP8266 microchip provides WiFi capability and can be programed with the Arduino IDE. Application examples for the MeteoMex aeria and terra kits demonstrate their functionality for air quality, soil, and climate monitoring. Further, a prototype for monitoring wastewater treatment is shown, which exemplifies the capabilities of the Wemos board for signal processing. The data are stored in a PostgreSQL database, which enables data mining. The MeteoMex IoT system is highly scalable and of low cost, which makes it suitable for deployment in agriculture 4.0, industries, and public areas. Circuit drawings, PCB layouts, and code examples are free to download from https://github.com/robert-winkler/MeteoMex.

## INTRODUCTION

The overuse of natural resources by humans and climate change have severe effects on the environment. As a consequence, the global food security is threatened, and pollution-related diseases such as allergies and asthma increase (*Vermeulen, Campbell & Ingram, 2012*; *Wheeler & Braun, 2013*; *D'Amato et al., 2015*; *Lake Iain et al., 2017*; *Cohen et al., 2017*; *D'Odorico et al., 2018*; *Dell'Angelo, Rulli & D'Odorico, 2018*; *Patella et al., 2018*).

In arid and semi-arid regions, irrigation is essential for agriculture. However, excess watering negatively affects food yield and quality (*El-Ansary, 2017*; *King, Stark & Neibling, 2020*) and leads to unnecessary consumption of water and energy. Besides, salt

Corresponding author
Robert Winkler,
robert.winkler@cinvestav.mx

accumulation in the soil reduces future yields (*Shrivastava & Kumar, 2015*; *Hutchinson, 2019*). The use of treated wastewater is possible for saving drinking water, but wastewater treatment requires energy for pumping and aeration and needs to be optimized (*Vergine et al., 2017*; *Miller-Robbie, Ramaswami & Amerasinghe, 2017*). Using greenhouses increases the productivity and quality in agriculture because lighting, ventilation, temperature, and watering can be adjusted to the cultivars and external conditions (*Jat, Singh & Kumar, 2020*).

A very efficient strategy to increase food availability is the protection of harvested products against insect infestation and spoilage. In tropical areas, losses of more than half of stored maize grains due to insect pests are possible. Adapted plant genotypes, agrochemicals, and improved storage strategies such as hermetically closed metal containers efficiently reduce insect-related postharvest losses (*López-Castillo et al., 2018*; *García-Lara, Saucedo Camarillo & Bergvinson, 2007*; *Tefera et al., 2011*). Besides, adequate storage conditions, that is, light, temperature, and humidity, prolong the shelf-life of human and animal food (*Bradford et al., 2018*). Some perishable food products require cooling during the complete production and distribution chain (*Mercier et al., 2017*). Cooling facilities—such as air conditioning, refrigerated trucks, fridges, and freezers—are primary drivers of industrial and domestic energy consumption, though (*She et al., 2018*).

Air temperature, relative humidity, and barometric pressure are main physical parameters for assessing weather and climate, and have direct effects on human health, ecology and agriculture (*Fagerlund et al., 2019*; *Villalobos et al., 2016*; *Yu, Qiao & Shi, 2018*; *Adejuwon & Agundiminegha, 2019*). Novel machine/deep learning algorithms for developing predictive weather and climate models rely on massive global datasets (*Dueben & Bauer, 2018*; *Scher, 2018*; *Racah et al., 2017*). On the other hand, local meteorological information is essential for evaluating microclimates and for optimizing farming (*Shock et al., 2016*; *Luwesi, Obando & Shisanya, 2017*). Air temperature and humidity also influence the emission of volatile organic compounds (VOC) from factories and building materials and the perceived indoor air quality (*Haghighat & De Bellis, 1998*; *Wolkoff, 1998*; *Fang, Clausen & Fanger, 1999*; *Milota & Lavery, 2003*; *Fechter, Englund & Lundin, 2006*; *Wolkoff & Kjærgaard, 2007*; *Liu et al., 2014*).

*Agriculture 4.0* is an umbrella term for using networked sensor data and artificial intelligence in food production. The collection of environmental data on Internet-of-Things (IoT) servers enables the development of complex predictive models. Besides, experts can provide services and give recommendations for remote locations.

Gathering highly localized data is crucial for the precise control of ideal plant growth conditions. *Smart farming* uses local soil data such as structure, composition, moisture, salinity, cation exchange capacity (CEC), and pH to optimize production (*Corwin et al., 2003*; *Grisso et al., 2005*; *Ould Ahmed, Inoue & Moritani, 2010*). One approach is remote sensing with satellites or aerial vehicles such as drones with hyper-/multispectral imaging (*Mulla, 2013*; *Saura, Reyes-Menendez & Palos-Sanchez, 2019*). For the continuous monitoring of environmental parameters, the installation of local sensors is common. A well-equipped greenhouse could contain hundreds of sensors that are connected to a central control unit (*Chaudhary, Nayse & Waghmare, 2011*).

Industrial farming already uses IoT systems for increasing productivity and efficiency. The European Union supports the development of IoT technology for the agricultural and food sector with the project *Internet of Food and Farm 2020* (https://www.iof2020.eu). However, for small stakeholders in developing countries, such commercial *agriculture 4.0* technology is usually out-of-reach, despite its tremendous potential in environmental protection and food production, especially in vulnerable regions (*Antony et al., 2020*; *Luthra et al., 2018*). Further, most industry-grade IoT systems are built on proprietary hardware and software and require specialists for their operation and adaptations.

Initiatives such as the *Public Lab* (https://publiclab.org) and *Lab On The Cheap* (https://www.labonthecheap.com) (*Gibney, 2016*), in contrast, promote the community-driven development of open technology. Such low-cost and *do-it-yourself* (DIY) devices are not only suitable for *crowd-sourcing* data in so-called *citizen science* (*Dickinson et al., 2012*), but also state-of-the-art research in the instrumental analysis (*Martínez-Jarquín et al., 2016*, *Rosas-Román et al., 2020*). Environmental sensing projects often use simple microcontroller boards such as Arduino (https://www.arduino.cc) and Wemos (https://www.wemos.cc) variants. The *Cave Pearl Data Logger* demonstrates that such devices can operate under harsh conditions (underwater) for more than 1 year on $3 \times AA$ battery power (*Beddows & Mallon, 2018*). The *Solar Powered WiFi Weather Station v 2.0* (https://www.instructables.com/id/Solar-Powered-WiFi-Weather-Station-V20/) uses a Wemos board and connects wirelessly to IoT platforms such as Blynk (https://blynk.io) and ThingSpeak (https://thingspeak.com), or an MQTT (Message Queuing Telemetry Transport, https://mqtt.org) broker. Many of such excellent community projects on environmental monitoring have been reported. However, reproducing DIY devices requires technical skills, and integrating the sensors into a professional IoT framework is too challenging for end-users. Remote training of farmers is possible (*Seelan et al., 2003*), but ideally the systems should be simple enough for being installed and operated by the local users with average education.

The MeteoMex project (http://www.meteomex.com) aims to unify the advantages of DIY and commercial systems and provides an IoT infrastructure for environmental monitoring in production and research with the following characteristics:

- **Scalable**. Printed circuit boards (PCB) and standard parts allow the mass production of identical sensing units. The database server can process thousands of operations per second.
- **Flexible**. The users can connect a huge variety of commercial sensors or integrate their prototypes.
- **User-friendly**. A simple design, pre-built modules, and code examples make the platform suitable for non-experts.
- **Low cost**. Generic electronic parts, the use of existing WiFi networks, and free software reduce the installation costs. The operation is economical because of low energy consumption and the possibility of self-hosting the IoT server.
- **Open**. All relevant hardware information and the software are completely documented and freely available.

## METHODS

### Microcontroller board, circuit, and PCB design

For connecting sensors to wireless networks, Arduino-compatible, single-board microcomputers are used. The low-end Wemos D1 mini board is based on the ESP-8266EX chip (80 MHz, 4 Mb flash memory) and provides WiFi. The board measures 34.2 mm × 25.6 mm and weighs 3 g. With a micro-USB connection, the Wemos D1 can be powered and programed (https://www.wemos.cc/). The board features 11 digital input/output pins and one analog input (3.2 V max.). The price of a standard Wemos D1 mini board is approximately 5 USD.

For mounting sensors and other electronic components, printed circuit boards (PCB) were designed that can be stacked on the Wemos boards-so-called *shields*.

The open-source software Fritzing (https://fritzing.org) was used for designing the circuits and printed circuit boards (PCB) (*Monk, 2015*). For fabrication, the PCB layouts were exported to the Gerber file format (https://www.ucamco.com/en/gerber). PCB figures in this article were produced with a Gerber file viewer (https://www.pcbgogo.com/GerberViewer.html). ALLPCB, China (https://www.allpcb.com/) produced testing lots of 100 pieces each.

The editable Fritzing files and example programs are available from the MeteoMex GitHub repository (https://github.com/robert-winkler/MeteoMex).

The configuration of the different shields is explained in the following subsections.

### MeteoMex aeria

The MeteoMex aeria (Fig. 1) shield uses a BME280 chip (Bosch, Stuttgart, Germany, https://www.bosch-sensortec.com/products/environmental-sensors/humidity-sensors-bme280/) to monitor the ambient air parameters temperature, relative humidity, and barometric pressure.

An additional CCS811 sensor (https://www.sciosense.com/products/environmental-sensors/ccs811-gas-sensor-solution/), permits the detection of total volatile organic compounds (VOC). Both chips use the $I^2C$ bus for sensor data transfer. If the board is powered by batteries, programming a *DeepSleep* mode is recommendable for reducing the energy consumption in idle mode. In *DeepSleep* mode, all functions except the real-time clock are switched off, reducing the power consumption of the ESP8266 chip from 70 mA to 20 µA (https://www.instructables.com/ESP8266-Pro-Tips/). For waking up the Wemos D1 mini, the jumper *J1* between RST and D0 (GPIO16) must be closed. To enable the *DeepSleep* function, therefore, a wire bridge has to be soldered into the jumper *J1*.

### MeteoMex terra

The MeteoMex terra (Fig. 2) shield is designed to connect an analog sensor, such as a conductive or capacitive soil moisture sensor, and a digital DS18B20 temperature sensor (Maxim Integrated). For operating the DS18B20 sensor, a pull-up resistor of 10 kΩ (*R1*) is required.

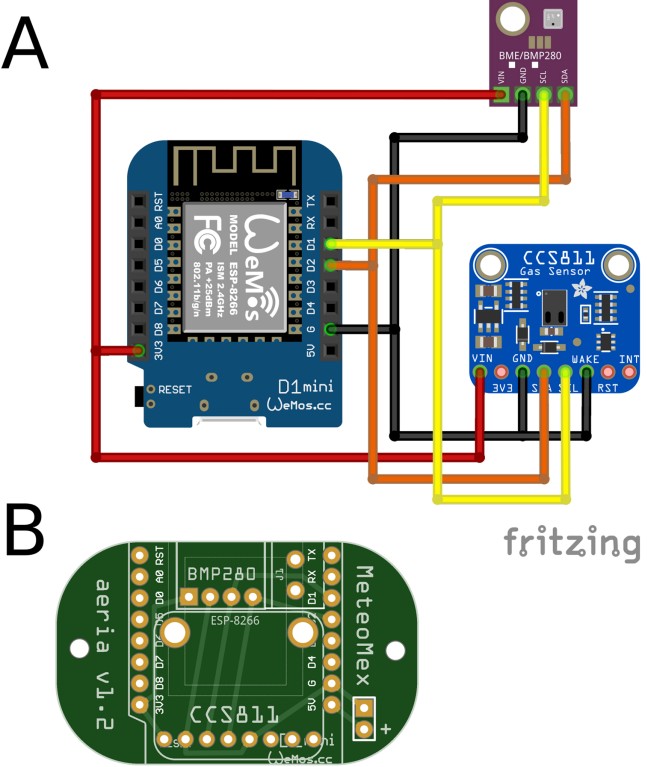

**Figure 1 MeteoMex aeria (A) circuit and (B) PCB shield.** The shield connects two sensors using the I2C bus: An BME280 sensor board for measuring temperature, relative humidity and a CCS811 sensor board for total volatile organic compounds (VOC). Closing jumper J1 is required for using DeepSleep/ WakeUp.

The power supply of the analog and digital sensor is connected to the pins D1 and D3, respectively. However, some analog sensors require a connection to the 5V pin for reliable operation. The maximal permitted input voltage for the A0 analog pin is 3.2 V. Thus, a voltage divider needs to be used when using sensors with higher signal levels.

This board as well provides a *J1* option.

### MeteoMex WasteWater prototype

The MeteoMex WasteWater (Fig. 3) configuration is a slight variation of the terra circuit. An analog Arduino turbidity sensor (TresD Print Tech) is used for measuring total suspended solids (TSS). For determining tank filling levels, a Jsn-sr04t waterproof ultrasonic sensor (Ranmex) is connected. Both sensors operate on 5 V. For this custom design, no PCB was printed, but the circuit was built on a Perfboard.

All boards were designed for the use of through-hole electronic components for facilitating manual assembly and soldering.

### Programming and Internet-of-Things infrastructure

An overview of the IoT infrastructure is shown in Fig. 4. The programming was done on Windows 10 and standard Linux distributions (Fedora and Ubuntu). The IoT platform is

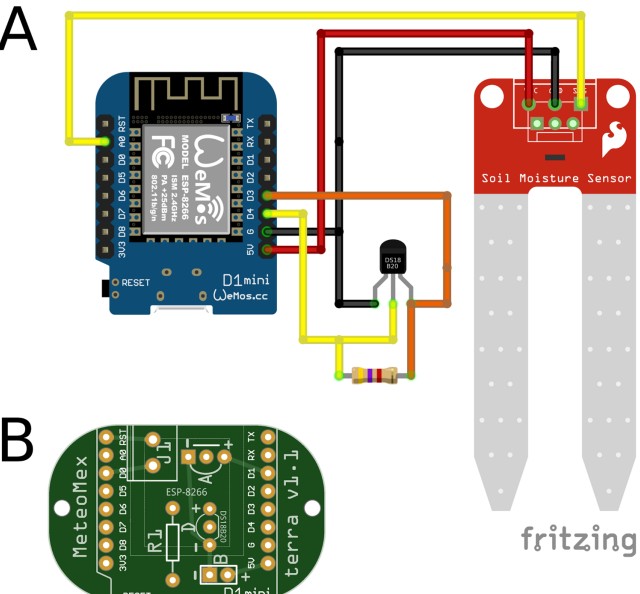

**Figure 2 MeteoMex terra (A) circuit and (B) PCB.** The connection of one analog soil moisture and one digital DS18B20 temperature sensor is possible. A 10 kΩ pull-up resistor is necessary for temperature measurement. The jumper J1 is used for programing DeepSleep/WakeUp.

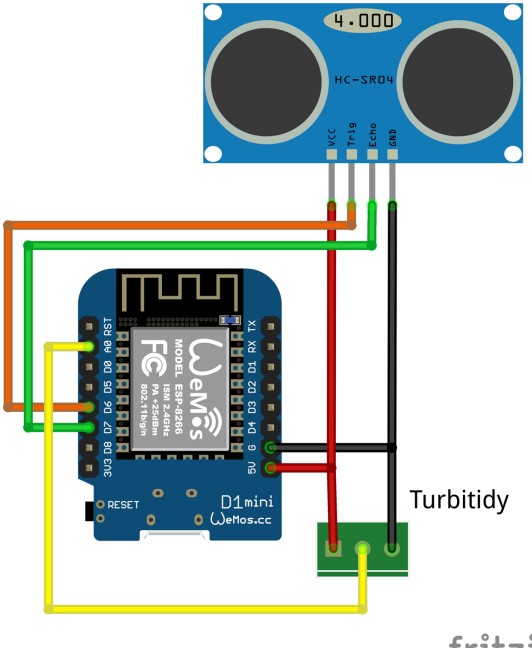

**Figure 3 MeteoMex WasteWater prototype circuit.** The ultrasonic SR04 sensor and the analog sensor for total suspended solids (TSS) both require a 5 V power supply.

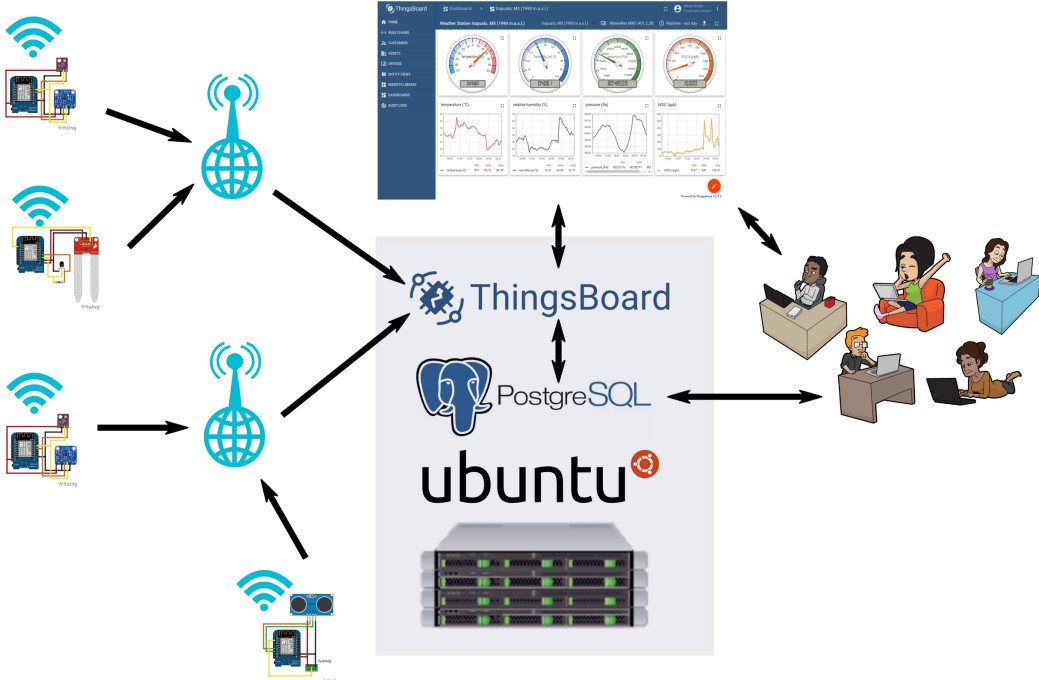

**Figure 4 Scheme of the MeteoMex IoT infrastructure.** The devices and the IoT server are located on different continents. Image credits: SVG Silh (WiFI: https://search.creativecommons.org/photos/59064244-d668-4e93-a508-565dfa372858, Antenna: https://search.creativecommons.org/photos/5481eacc-ec08-49ee-8e77-7f6564337631, Globe: https://search.creativecommons.org/photos/3114893d-208a-4429-b53c-dadf66356260), CC0 1.0 Universal (CC0 1.0) Public Domain Dedication; Server: Wikimedia Commons (https://search.creativecommons.org/photos/f64cfdc1-ff82-49ad-9f72-af98182ccf61), Attribution 4.0 International (CC BY 4.0); ThingsBoard: ThingsBoard, https://thingsboard.io/mediakit/, © 2021 The ThingsBoard Authors; Ubuntu: Canonical Ltd. (https://en.wikipedia.org/wiki/Ubuntu#/media/File:Logo-ubuntu_no®-black_orange-hex.svg), Public Domain; PostgreSQL: Daniel Lundin (https://en.wikipedia.org/wiki/PostgreSQL#/media/File:Postgresql_elephant.svg), PostgreSQL License.

running on a Virtual-Private-Server, hosted by IONOS (https://www.ionos.de). The operating system of the IoT server is Ubuntu 18.04 LTS (https://ubuntu.com).

### Programming the Wemos board

The Wemos boards are Arduino-compatible and can be programed with the open-source and cross-platform software Arduino IDE (https://www.arduino.cc/en/Main/Software), using the additional ESP32 board definitions from https://dl.espressif.com/dl/package_esp32_index.json. Compiled programs were transferred to the Wemos boards using a USB interface. For testing and debugging, a serial monitor window was used.

### Internet-of-things platform and database

For registration and administration of devices, collecting telemetry data, and visualization, the *Community Edition* of the open-source IoT Platform ThingsBoard (https://thingsboard.io) was used.

All device and telemetry data are stored in a PostgreSQL database (https://www.postgresql.org). For database queries and administration, the Adminer (https://www.adminer.org)

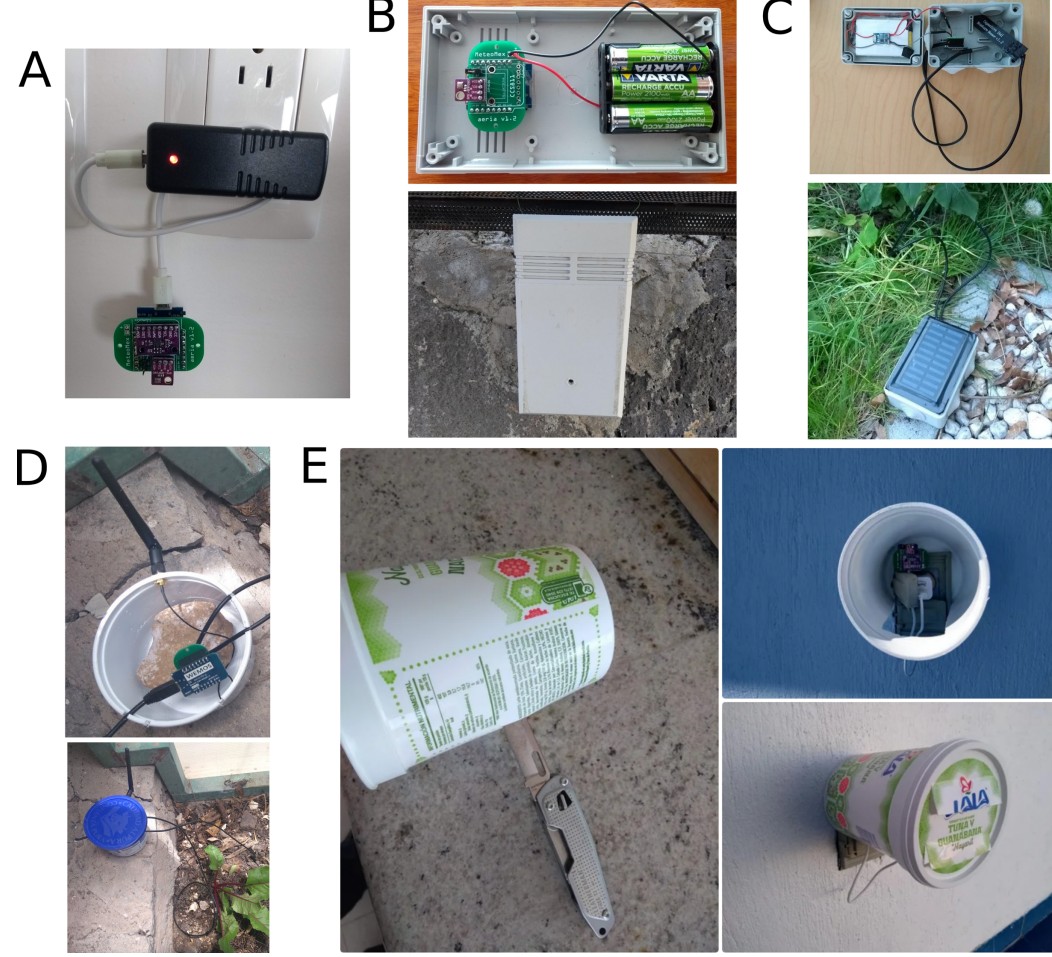

**Figure 5** Different housing and power supply options: (A) no housing and direct powering with USB charger, (B) 3 × AA rechargeable batteries in commercial enclosure, (C) solar panel and lithium battery with wet room installation box, (D) and (E) re-use of plastic beakers.

database management web interface was installed. Regular maintenance tasks are performed with SQL scripts and server CRON jobs.

### Housing and power supply

MeteoMex kits provide no housing by default, which saves costs and reduces unnecessary plastics waste. The circuits operate at low voltages (5 V), so the devices are safe for humans and animals. For indoor air monitoring, the devices can be simply connected to a USB power supply (Fig. 5A). For outdoor conditions, the protection against dust, insects, and water might be necessary (Figs. 5B–5E). However, re-used plastic beakers, for example, from dairy products, usually fulfill this purpose.

Alternatively, to USB port power, the devices can be operated with 3 × A.A. rechargeable batteries or with solar panels and a power bank. A MeteoMex aeria device with BME280 sensor works about 3 months with 3 × AA rechargeable batteries, with hourly measurement and using a *DeepSleep/WakeUp* routine. However, additional

electronic components and batteries have a negative impact on the system's environmental footprint.

The building of the device and housing shown in Fig. 5E is described in an Instructable (https://www.instructables.com/member/RobertWinkler/instructables/).

### Availability of MeteoMex kits and code

Additional documentation for building and programing the devices and kits are available from the MeteoMex project page http://www.meteomex.com. PCB Fritzing layouts and code examples are deposited at the GitHub repository (https://github.com/robert-winkler/MeteoMex) with open license terms. The code described in this paper was archived as release v1.1 at Zenodo (http://doi.org/10.5281/zenodo.4075278, *Winkler, 2020*).

## RESULTS

### Internet-of-Things integration of MeteoMex units

According to the legend, Internet-of-Things (IoT) started in the early 1980s, when a Coke-vending machine at the Carnegie Mellon University was connected to the internet (https://www.cs.cmu.edu/~coke/). In these pioneering times, integrating custom hardware into a computer network was a challenge for technical specialists. Nowadays, different IoT standards and protocols facilitate the data telemetry and processing (*Ponnusamy & Rajagopalan, 2018*).

The IoT platform ThingsBoard supports various telemetry protocols. For the MeteoMex project, HTML was chosen for its simplicity in programing and testing. The correct set-up of a device can be verified by sending data from a computer console. For example, the following command sends the value 99 of the variable humidity for a device with the token uVCGuzjBqV:

```
curl -v -X POST -d "{\"humidity\":99}"
   http://www.meteomex.com:8080/api/v1/uVCGuzjBqV/telemetry
   --header "Content-Type:application/json"
```

Wemos boards also could be connected to other IoT platforms. The integration into Blynk (https://blynk.io) and Thinger.io (https://thinger.io) was successful. But apart from technical questions, the licensing scheme of IoT platforms is relevant. ThingsBoard (https://thingsboard.io) is licensed under the Apache 2.0 license (https://www.apache.org/licenses/LICENSE-2.0). That is, besides being open-source, ThingsBoard offers a free *Community Edition* and allows its employment in commercial applications. ThingsBoard is cross-platform compatible and can be installed on Windows, Mac, and Linux (https://thingsboard.io/docs/user-guide/install/installation-options/). The system used in this study was installed on Ubuntu 18.04 LTS (https://ubuntu.com). The computational capacity of the Virtual-Private-Server, hosted by IONOS (https://www.ionos.de), can be adjusted to the IoT server load. However, the installation of a ThingsBoard *Community Edition* IoT server is possible at no software cost.

The open-source PostgreSQL (https://www.postgresql.org) database server for storing the IoT data has high performance and robustness. Database maintenance and data

manipulation are possible with system tools and external programs. Exported data can be further analyzed, for example, with statistics and data mining software, such as R/Rattle (https://rattle.togaware.com) (*Williams, 2011*).

Transferring data with radio frequency (R.F.) and Bluetooth was tried as well. However, the direct connection of the devices to WiFi networks turned out the technically easiest solution. WiFi networks are ubiquitously available, of fair security, and no additional adaptors are necessary to send collected data to the internet.

The ThingsBoard dashboards are visualized on a standard web browser, making it compatible with standard personal computers and mobile devices. The web platform also permits the setting-up of data processing pipelines, the *IoT Rule Engine*, and the definition of alarm levels and actions.

At the time of writing this manuscript, the IoT server was running for more than 180 days without interruption, demonstrating technical robustness. Although the IoT server was located in Germany, and the sensor units in Mexico, no telemetry data transfer problems were noticeable. Local power cuts or internet failures only affect devices in a particular zone. Since the wireless network settings are saved to the ESP8266 flash memory, the Wemos boards reconnect when rebooting.

Thus, the overall infrastructure takes into account the main aspects of an IoT system for agriculture (*Elijah et al., 2018*), such as cost, simple and robust technology, localization, scalability, and interoperability. Open licenses of hardware and software, and the use of common standards (WiFi network, HTML telemetry, SQL database) assure a long-term, cost-efficient, and provider-independent service.

Different application examples are presented in the next part.

## Example 1: weather station for climate and volatile organic compounds

Climatic conditions and air pollution directly affect human health and welfare. Therefore, public monitoring data inform the citizens about possibly hazardous levels of contamination.

Figure 6 shows the recording of climate and total volatile organic compounds (tVOC) of a MeteoMex aeria device installed outdoor at the *National Institute of Respiratory Diseases*, INER, Mexico City, Mexico (http://iner.salud.gob.mx/). The apparatus is installed next to an aerobiology station for the continuous monitoring of pollen and microbial spores.

The readings show a daily tVOC peak at about 8 a.m., which could be caused, for example, by the morning traffic. Sensitive persons should avoid physical activities in periods of increased air contamination. Widely distributed public sampling stations could provide more localized data for warning people about possible health risks. The information also could motivate to reduce contaminating activities in affected areas.

The climate parameters temperature, humidity, and pressure, which are simultaneously recorded, also could be used to estimate the daily global solar radiation using a neuronal network model (*Jimenez et al., 2016*). $I^2C$ capable sensors for measuring

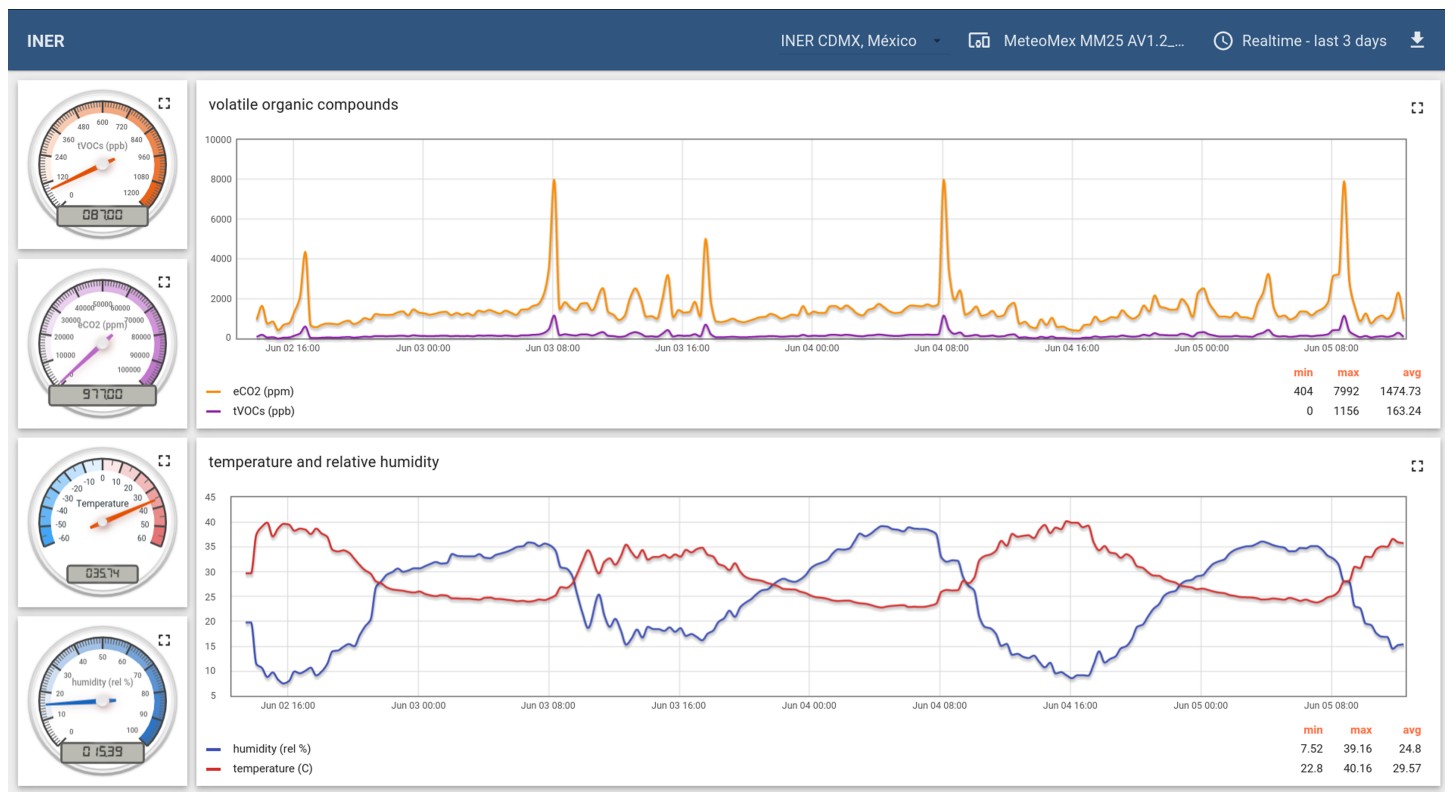

**Figure 6 Monitoring ambient air at the National Institute for Respiratory Diseases, INER, Mexico City, Mexico.**

photosynthetically active radiation (PAR) or Lux sensors could provide additional information about radiation and light conditions.

## Example 2: greenhouse monitoring (air and soil)

For the optimal growth of plants, adequate air and soil conditions are essential. The conductivity and the dielectric properties of soil depend on its composition, structure, moisture, and salinity (*Sreenivas, Venkataratnam & Rao, 1995*; *Malicki & Walczak, 1999*; *Wang & Schmugge, 1980*). Those can be easily measured, and it was shown that soil electrical conductivity measurements correlate with Local yield (*Grisso et al., 2005*).

The MeteoMex terra device uses either conductive or capacitive probes to estimate the soil moisture. The capacitive probes are protected against corrosion and, therefore, preferable. Since the analogous measurement only provides an integer value between 0 and 1,024, the signal needs to be calibrated. The most straightforward procedure is measuring the dry sensor's output and the sensor when completely submerged in water. A more realistic calibration is possible by adding water to dry soil until saturation. The soil type is crucial for the water storage capacity and the conductivity/ dielectric properties. Another strategy is the acquisition of raw readings and subsequent data interpretation.

Figure 7 shows on the left side, the air data, and on the right side, the soil data, which are measured hourly in a domestic greenhouse with automated irrigation. The capacitive soil

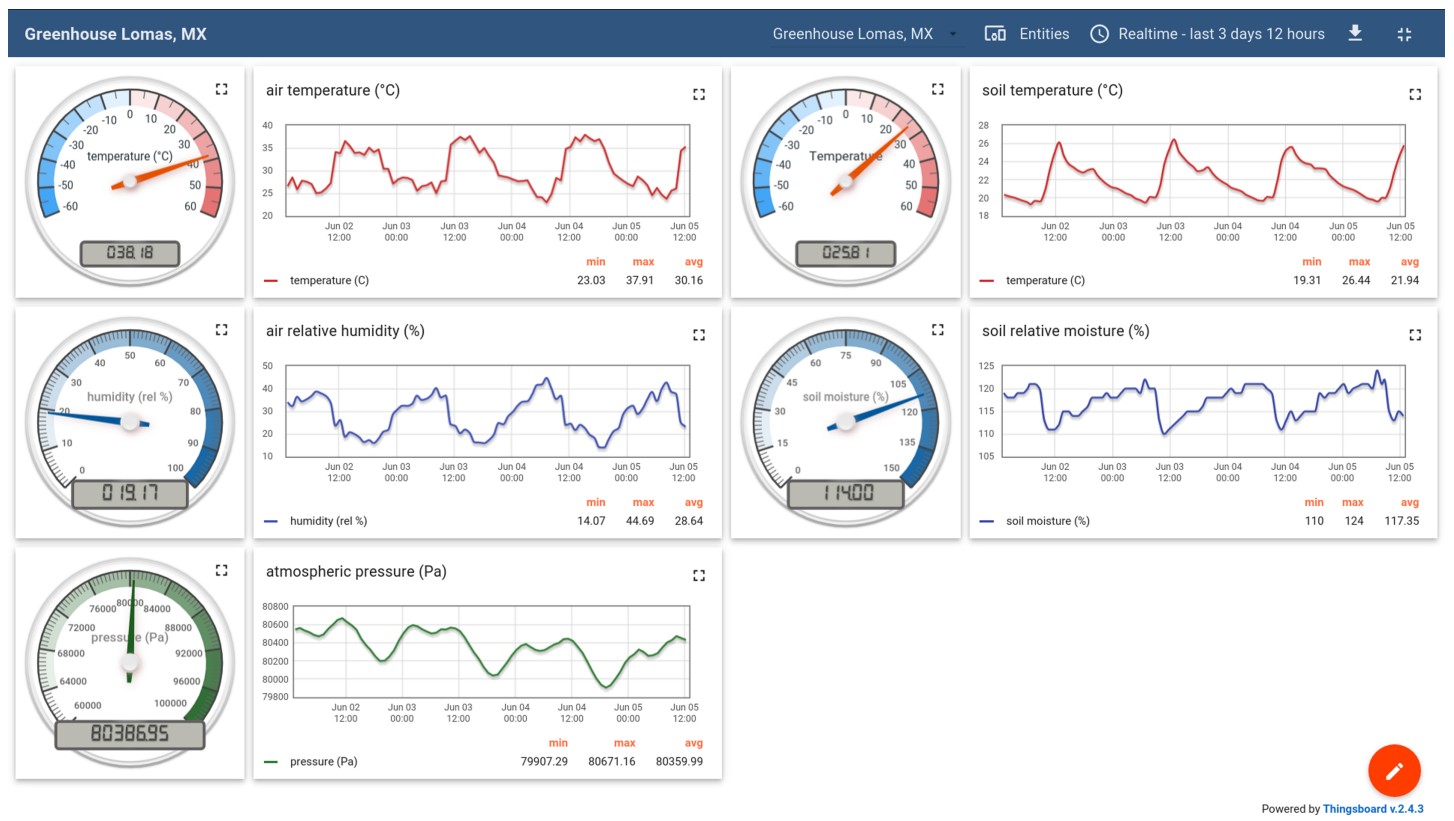

**Figure 7 Monitoring the air and soil parameters in a domestic greenhouse.**

moisture sensor was calibrated to 0–100% with purified water. The actual sensor readings range between 110% and 124%, which reflects the dielectric properties of soil.

The average soil temperature of about 22 °C is ~8 K lower than the air temperature, demonstrating the heat capacity and temperature buffer properties of the soil. Besides, the temperature variations during the day are ~15 K for air and ~7 K in the soil.

The low average barometric pressure of 80,360 Pa is consistent with the theoretical value of 80,572 Pa, which was calculated for an ambient temperature of 30 °C and an altitude of 1,990 m above sea level (https://www.mide.com/air-pressure-at-altitude-calculator).

As the temperature peak at about 12 p.m. with the following decline indicates, the measuring point is only exposed to direct sunlight for a short period of day. Temperature and solar radiation profiles have a direct effect on plant growth. Therefore, *micro-climate engineering*, for example, by planting shading trees, could become a common practice in future agriculture (*Trilnick, Gordon & Zilberman, 2018*).

The well-being of farm animals also depends on climate conditions. Temperature and humidity measurements in the barn demonstrated that the heat stress of dairy cattle in Canada was underestimated when using data from meteorological stations. Therefore, reading environmental data in the barn is recommended to determine the actual conditions the cows are exposed (*Shock et al., 2016*).

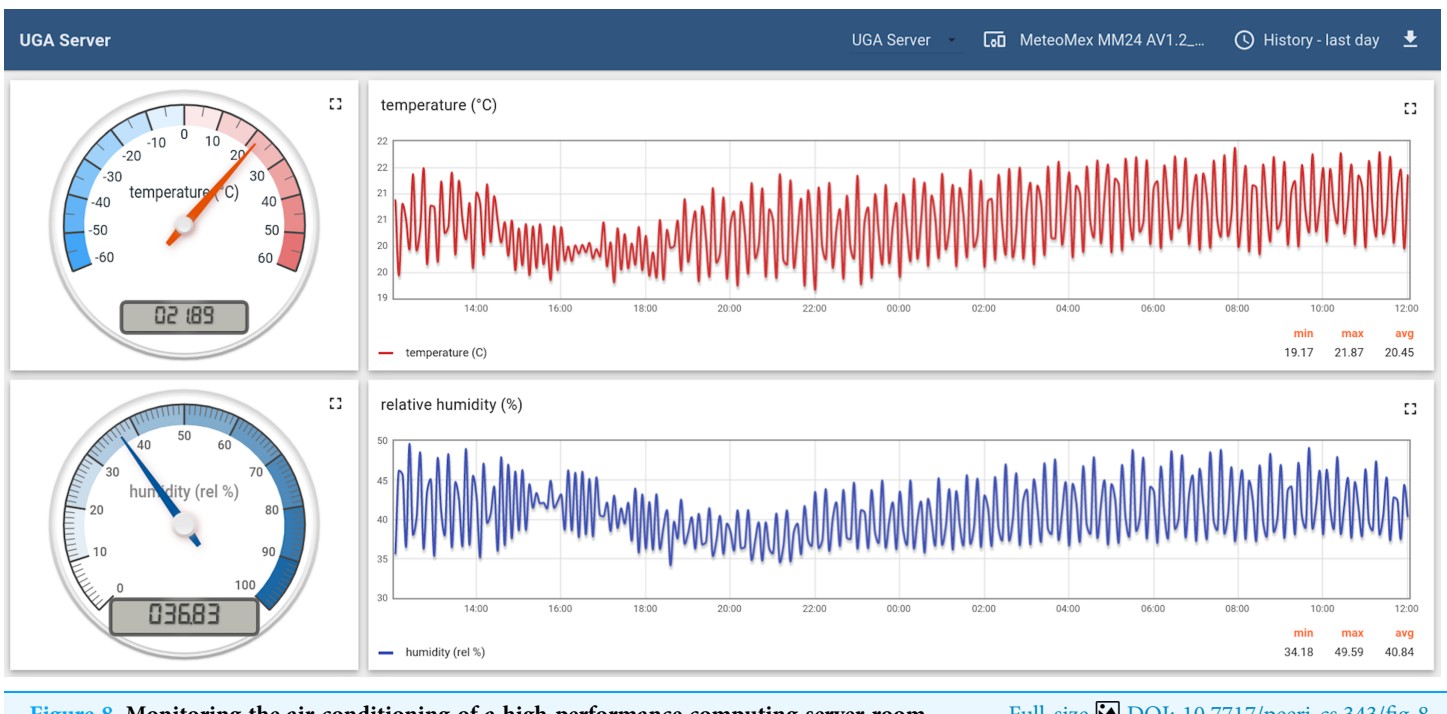

**Figure 8** Monitoring the air conditioning of a high-performance computing server room.

The economical sensors and the open IoT platform enable numerous applications in *agriculture 4.0*, *smart farming*, and *urban greening* (*Madushanki et al., 2019*).

## Example 3: high-performance computing room monitoring (air conditioning)

Delicate equipment such as scientific instruments and high-precision machines require controlled ambient conditions. Overheating or condensation could result in serious device damages. Figure 8 shows the monitoring of a high-performance computing server room at the *National Laboratory of Genomics for Biodiversity*, Cinvestav Irapuato, Mexico (https://langebio.cinvestav.mx). The charts demonstrate a tight temperature control with an average temperature of 20.45 °C and less than 1.5 K variation.

The constant monitoring of ambient conditions also helps to detect potentials for energy savings, such as changing the settings of an air conditioning system at night, or during periods of inactivity.

## Example 4: domestic wastewater plant

Using treated wastewater for irrigation saves sweet water reserves in dry areas. Figure 9 shows the block chart of a domestic wastewater treatment plant. The *activated sludge process* for reducing organic soluble solids by microbes requires aeration. This aeration step and the pumping for filtration and irrigation require electric energy. The *food-energy-water nexus* describes the strong interconnection between these resources (*D'Odorico et al., 2018*).

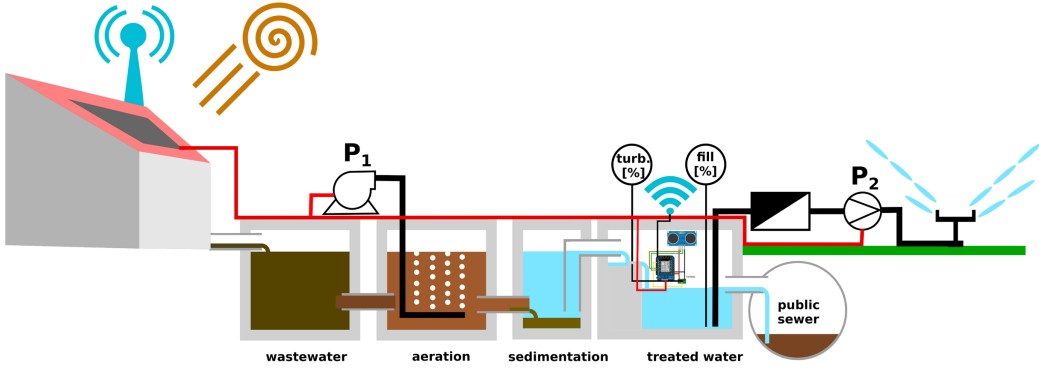

**Figure 9 Monitoring tank filling level and turbidity in a domestic wastewater plant.**

The presented facility uses photovoltaic energy. For saving electricity, the aeration is not operated continuously, but only during the daytime, and in half-hour intervals. The quality of the treated water needs to monitored to avoid either over-purification (wasting energy) or insufficient purification (clogging filters). In addition, the irrigation scheme has to be adjusted to the generation of treated water. Unused treated water passes through an overflow-pipe to the public sewer and is lost for irrigation.

The MeteoMex WasteWater prototype (Figs. 3 and 10) has two sensors: An analog Arduino turbidity sensor for measuring total suspended solids (TSS), and an ultrasonic distance sensor for estimating the tank filling. The Wemos board and the sensors are located inside the wastewater treatment plant, which is built from ferroconcrete. Thus, a direct WiFi connection to the wireless network is not possible. Thus, a Wemos D1 mini Pro board with an external antenna was used.

The data of both sensors are noisy (Fig. 11). Air bubbles and particles can create spikes in the turbidity measurements. Scattering and sometimes erratic readings of the ultrasonic distance sensor are also common and need to be addressed in the data processing. Two methods to filter noisy signal data are presented here.

### Chopping off turbidity spikes

In the case of the turbidity sensor, spikes, that is, individual values with high apparent intensity, should be removed. This was done using a circular buffer, which was set-up in the variable declaration section of the Arduino program:

```
// Circular buffer set-up
//<https://github.com/rlogiacco/CircularBuffer>
#include <CircularBuffer.h>

CircularBuffer<float,10> turbbuffer;
// circular buffer capacity for turbidity is 10
```

In the program loop, the minimum value of the last ten readings is determined:

```
//Turbidity
int turbSensor = analogRead(17);
```

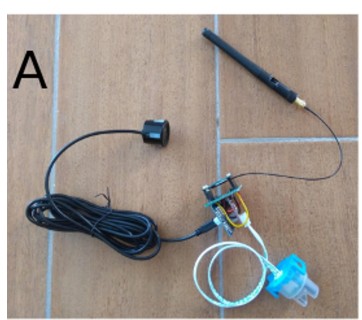
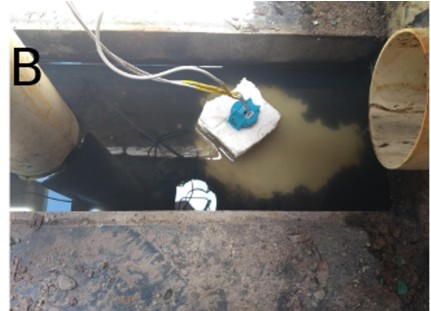
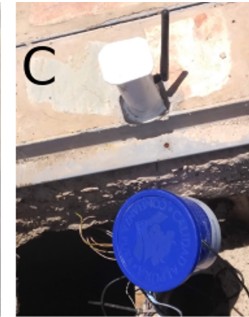
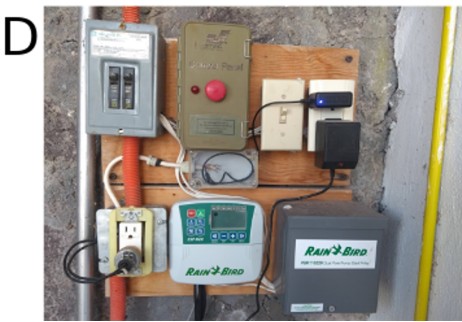
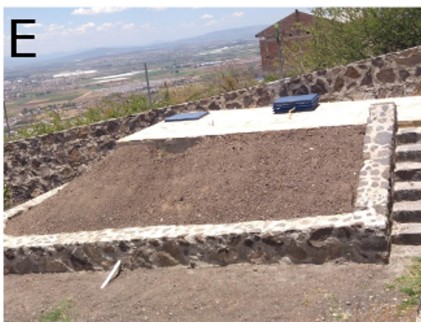

**Figure 10 Installation of the MeteoMex WasteWater prototype in the domestic wastewater plant.**
(A) Connection of the sensors to the custom-built Wemos shield, (B) the turbidity sensor is stuck into a piece of styrofoam for floating, (C) protection of the Wemos board against rainwater by a plastic beaker, and external antenna, (D) control board for wastewater treatment and irrigation; the power supply with blue light provides the energy for the Wemos board, (E) outside view of the wastewater plant.

```
float turbpercent = map(turbSensor,1024,0,0,100);

turbbuffer.push(turbpercent);

float turbpercentmin = 100;
using index_t = decltype(turbbuffer)::index_t;
for (index_t i = 0; i < turbbuffer.size(); i++) {
  if (turbbuffer[i] < turbpercentmin) {
      turbpercentmin = turbbuffer[i];}
}
```

The reported turbidity is slightly underestimating the real value, and the measurement is more sluggish than using the raw readings. However, the spikes are removed efficiently, and the filtered data are more robust, which is important, for example, for setting an automated alarm level.

### Kalman filter for tank level readings

Correcting the readings of the ultrasonic sensor is more complex since random deviations can be in both directions. Thus a Kalman–Filter was used for minimizing estimation errors (*Kalman, 1960*; *Simon, 2001*).
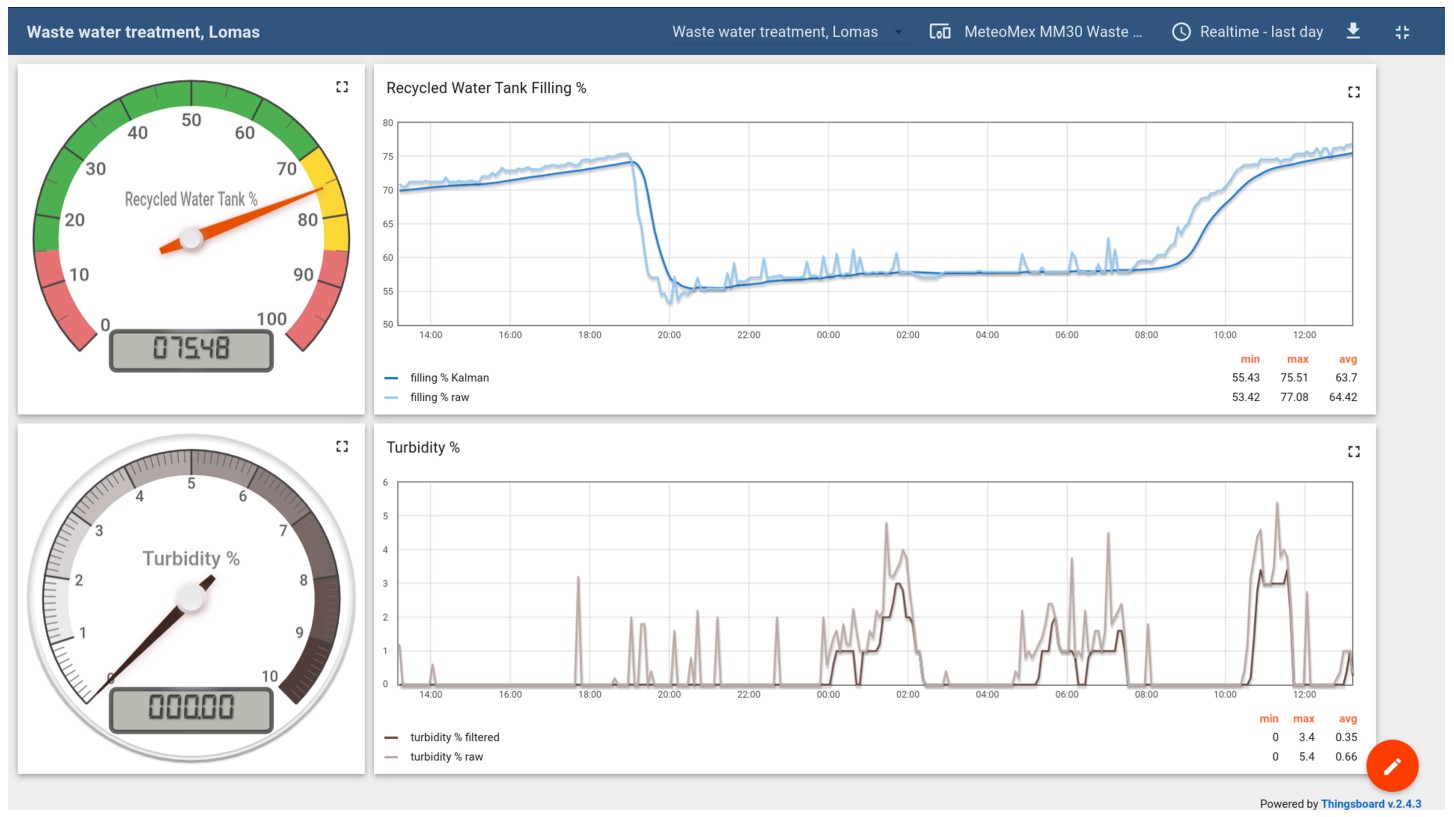

**Figure 11  Monitoring tank filling level and turbidity in a domestic waste water plant.** Raw sensor data were processed by the Wemos ESP8266 processor to remove noise.               

The respective Arduino library is included in the variable declaration section:

```
// Kalman Filter library
//<https://github.com/denyssene/SimpleKalmanFilter>
#include <SimpleKalmanFilter.h>
SimpleKalmanFilter ultrasonicKalmanFilter(1, 1, 0.01);
```

The filling level is calculated from the reading of the ultrasonic sensor SR04 and the tank dimensions (for details see the program in the GitHub repository).

```
float fillpercent = 100 * (1-((distance-47.5)/120));
```

```
// apply Kalman filter
float fill_estimate = ultrasonicKalmanFilter.updateEstimate
(fillpercent);
```

Although the maths behind the Kalman filter is not trivial, it could be easily implemented, and the performance of the ESP8266 is sufficient for real-time signal processing.

## DISCUSSION

### Sustainability and socio-economic impact

As for any technology, the possible negative aspects of its adoption need to be discussed. IoT devices consume energy for themselves and for the data transfer and processing infrastructure. Further, their production consumes resources, and at the end of their lifetime, they generate electronic waste (https://whatis5g.info/). Additional environmental issues arise when the devices are powered with batteries.

On the other side, the IoT technology can significantly contribute to saving natural resources (see examples above). The energy consumption of the presented domestic wastewater treatment plant (Example 4, "Example 4: Domestic Wastewater Plant") is several kWh/day, compared to only some Wh/day of the MeteoMex device. The energy demand of the IoT equipment corresponds to about 0.1% and enables the continuous monitoring and optimization of the plant. Even little improvements in the wastewater treatment and irrigation process that reduce the energy consumption by a few percent justify the environmental and economic cost of the IoT integration.

In other cases, the IoT devices could be mobile and only connected for project-specific tasks, for example, for determining the day-night temperature profile of a production facility.

Sustainability was a central design goal of the complete platform. The used boards are highly integrated and provide the necessary computation and networking functions with a minimum of material (3 g for a complete Wemos D1 mini board) and energy. WiFi technology and 5 V USB power supplies are globally available, and no special adapters are necessary.

Importantly, the low cost of the hardware components and the permissive licenses for all parts of the infrastructure—circuit and PCB board design, database, and IoT platform—make the adoption of an IoT system in marginal production sites feasible. Further, weather enthusiasts and environmental activists could form networks for regional and global data collection. The program code can be re-used by the community, for example, to integrate more sensors, and for education.

## CONCLUSIONS

The MeteoMex project aims towards a community-driven Internet-of-Things (IoT) framework. Despite the use of basic hardware components and free software, the infrastructure reaches professional-grade performance and robustness.

Monitoring the environmental parameters helps to protect natural resources (water and energy), to timely detect health-hazards, and to increase the production of high-quality food. New, data-driven strategies for food production, such as *micro-climate engineering*, *smart farming*, and *precision agriculture*—here summarized as *agriculture 4.0* require highly localized data. Collecting the readings of multiple simple sensors could provide more useful information than high-resolution data from sparse measurement stations. The presented IoT infrastructure is highly scalable and can process telemetry data from few to thousands of sensor units.

An essential characteristic of this IoT system is the availability of electronic circuit designs and PCB layouts, program codes, and software under open-source licenses. Further, existing infrastructure such as WiFi networks is used to improve the economic and environmental sustainability. The IoT users are not *locked in* within a proprietary technology, but free to choose from multiple vendors, if they need replacement parts or technical service. The comprehensive documentation and the availability of PCB shields for frequently needed set-ups facilitate the do-it-yourself (DIY) assembly of IoT units. Additional sensors can be easily integrated due to the flexible and modular design of hardware and software.

## ACKNOWLEDGEMENTS

I thank the engineers of ThingsBoard for technical advice, and Dr. Cei Abreu (*National Laboratory of Genomics for Biodiversity*, UGA/Langebio, Cinvestav Irapuato, Mexico) and Dr. Josaphat Montero-Vargas (*National Institute for Respiratory Diseases*, INER, Mexico City, Mexico) for providing sensor data. Equipment and material for the study were provided by Kuturabi S.A. de C.V.

### Funding

Equipment and material for the study were provided by Kuturabi S.A. de C.V. The funders had no role in study design, data collection and analysis, decision to publish, or preparation of the manuscript.

### Grant Disclosures

The following grant information was disclosed by the authors:
Kuturabi S.A. de C.V.

### Competing Interests

Robert Winkler is an Academic Editor of PeerJ and a shareholder of the company Kuturabi S.A. de C.V.

### Author Contributions

- Robert Winkler conceived and designed the experiments, performed the experiments, analyzed the data, performed the computation work, prepared figures and/or tables, authored or reviewed drafts of the paper, and approved the final draft.

### Data Availability

Fritzing files (Circuit and PCB design) and Arduino code examples are available at GitHub: https://github.com/robert-winkler/MeteoMex.

Further documentation and kits are available from the MeteoMex project page http://www.meteomex.com.

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
