# Peer review of "MeteoMex: open infrastructure for networked environmental monitoring and agriculture 4.0"

_PeerJ Computer Science, doi:10.7717/peerj-cs.343_

## Round 0.1 · original submission · Major Revisions

This work is interesting and novel but requires improvements to justify publication. Address each reviewer comment carefully, particularly those comments from Reviewer 2.

Reviewer 1 ·

Basic reporting

The article should include a section named Background to demonstrate how the work fits into the broader field of knowledge.
Relevant prior literature must be referenced, for example:
Mulla, D. J. (2013). Twenty five years of remote sensing in precision agriculture: Key advances and remaining knowledge gaps. Biosystems engineering, 114(4), 358-371.
Saura, J. R., Reyes-Menendez, A., & Palos-Sanchez, P. (2019). Mapping multispectral Digital Images using a Cloud Computing software: applications from UAV images. Heliyon, 5(2), e01277.
Seelan, S. K., Laguette, S., Casady, G. M., & Seielstad, G. A. (2003). Remote sensing applications for precision agriculture: A learning community approach. Remote sensing of environment, 88(1-2), 157-169.

Experimental design

The research work is clearly defined. It is relevant and meaningful.

Validity of the findings

The investigation was conducted rigorously but the author should improve the conclusion.

Reviewer 2 ·

Basic reporting

1) The paper has several strong points, as it has a good literature review and is very didactic about the proposed system implementation. However, there is one major problem as there is no scientific contribution.

The article describes a technical solution for a common problem in smart farms, although it uses and describes commercial hardware modules and free software. Thus, there is no new technique, hypothesis or study.

This is notable in the text, as there is no problem stated, no highlights of the paper scientific contribution and its main objective.

2) The introduction does provide sufficient background and include several relevant references. It is one of the strongest points of the paper. Most of references are from the past 5 years and there is an excellent review of them. However, I would recommend avoiding some parts as L49-53 (perishable food) and L60-64 (factories examples), they are a bit off the paper main topic. An explanation of smart farms and its context would be great before L65. In addition, the introduction seems to be missing an ending paragraph to conclude the idea and relate to the paper’s main topic.

3) English language and style are fine, with just a few minor spell checks required. However, some paragraphs are fragmented, thus the text needs a small revision (For example, L46-47).

4) Most of figures are good with enough information. One exception is the text in figures 6, 7, 8 and 11, which is not legible in the pdf.

Experimental design

It is an interesting topic and matches the paper scope. Although it isn’t an easy implementation and requires a lot of study and tests, there is no relevant and meaningful scientific contribution. It is a technical solution for a problem.

A solution would be to make a comparison study with other solutions and actually test its performance or propose a new structure in communication. There are several solutions for this application with better technologies, using Zigbee and Lora networks (for example). However, it would still be hard to find a scientific gap for the study.

My suggestion for the author is to change the paper scope/type to a literature review. The paper review is excellent and updated. There aren’t many reviews about these systems lately and would be great with your example of implementation with MeteoMex.

For this, only a few things have to change. For example, an expansion of the literature review as a “first part”, an example of sensor module construction with MeteoMex for a second part and some “study cases” as a third part.

Validity of the findings

There is no finding in the paper, although it is an excellent description of a technical solution with free software and accessible hardware.

Additional comments

I believe the paper is not suitable for publication as it is now. However, I suggest to turn it to a literature review paper as I described in 2.

·

Basic reporting

No comment

Experimental design

No comment

Validity of the findings

No comment

Additional comments

The MeteoMex project appears to be a fairly complete set of design files and software examples necessary for making a functional sensor reporting device. I have raised some issues in the specific comments below that I feel would make the manuscript more readable and useful to the interested reader. While the files are made available on Github (a commercial site with an uncertain long-term existence), I would recommend that you also create a permanent archived copy of the existing repository on a long-term archiving site such as zenodo.org, which provides free archiving. It’s certainly appropriate to provide updated files on Github on an ongoing basis.
The introduction could be more concise, as the listing of the various challenges of food production, storage, wastewater treatment, etc. seem to go on for a long time before the crux of the project is reached, which is a system to help monitor various aspects of these various industries and environments.
Line 19 Abstract: “analog, digital and I2C sensors” should be reworded, since in this context I2C is a digital protocol and a sensor using the I2C protocol is not different from the generic term “digital sensor”.
Line 46-47: This is a sentence fragment
Line 57-58: This sentence doesn’t seem relevant to the thrust of the paper. It could be removed.
Line 73: agriculture 4.0 is used here without a useful definition. The only definition I find appears at the end of the paper at line 380.
Line 86: AA battery does not need periods after each A.
Line 88-90: It’s not immediately clear what the purpose or service provided by Blynk, ThingSpeak, MQTT is, for a reader not already familiar with those services. This sentence should be split into two, and the role of these services should be more clearly explained. The MQTT acronym should be defined.
Line 90-91: This sentence conveys no useful information and could be cut to make the introduction more concise.
Line 93-94: This sentence gives a brief description of the attributes of the MeteoMex project, but no description of the specific capabilities (i.e. what can/could it measure, how frequently, what kind of power requirements). Those capabilities are described in more detail in section 2, but a brief rundown should be provided in the introduction.
Figure 1: Which part the term “shield” is referring to in the caption isn’t self-evident until the reader has read the text and figured out that the PCB referred to in the caption is the shield. I would suggest changing the initial sentence of the caption to read “A) circuit and B) PCB “shield”.”
Line 99: The units of the board dimensions are given as mm squared, but the number provided are the linear dimensions (mm), so the mm2 label should be changed to mm.
Line 120-121: The DeepSleep mode is mentioned a few times in the manuscript, but it is never properly explained what this mode does. The sentence on Line 121 confuses me, since the way it is written makes it sound like the jumper needs to be closed manually in order to wake the device, and that this would need to be closed by the user each time they wanted to wake the device. But normally I would assume that a deep sleep mode would have some facility to automatically wake itself, either on a regular time interval or because of some external sensor interrupt being triggered, rather than the user needed to physically close a jumper. If the deep sleep mode works normally, this sentence should be reworded to say something like “For enabling the DeepSleep mode, the jumper J1 must be closed.”
Line 124: The manufacturer (Maxim) for the DS18B20 should be listed.
Line 127: From what I read, the maximum input voltage for the analog input on the Wemos is 3.2V, so it might be worth mentioning that analog sensors that use a 5V supply must still limit their output range from 0 to 3.2V to avoid damaging the analog channel on the microcontroller.
Line 131: The manufacturer and part number for the turbidity sensor should be provided.
Line 132: The manufacturer for the JSN-SR04T sensor should be provided.
Line 158: change “why” to “so”
Line 164-165: Awkward phrasing for the sentence beginning “Programming a DeepSleep/WakeUp…”
Line 166: What is an accumulator in this context? A battery?
Line 239: Using a neural network to estimate solar irradiance from temperature humidity and pressure seems overly complicated and not relevant to this project. A lux or PAR sensor (though PAR sensors are expensive) could give similar information on irradiance and also interface to the MeteoMex.
Lines 247-253: The process of calibrating a soil moisture sensor is glossed over here. Please provide a reference to give a more detailed description of a proper calibration routine.
Line 254-256: After giving the soil temperature in Celsius, the units change to Kelvin for the subsequent temperature values. Why not continue using Celsius, especially if they are interchangeable with respect to numeric value?
Line 260: I assume the temperature peak was at 12 pm, not am.
Line 261: The phrasing of this sentence is awkward. It would read better as “… point is only exposed to direct sunlight for a short period of the day.”
Line 270: “economical” instead of “economic”
Lines 272-281: I’m not sure that the example of monitoring air temperature a server room merits a separate section and figure here. Monitoring air temperature was adequately covered in section 3.3. I think most of section 3.4 could be cut to help make the paper more concise, and a mention of the server room air temperature results could be made in a sentence in the paragraph on lines 265-269 where another air temperature monitoring usage is described.
Line 329: change “importing” to “important”

---

## Round 0.2 · Minor Revisions

Dear author, although most of the comments from reviewer 1 and 3 were addressed, reviewer 2 still has some concerns about your improvements. Address all their comments carefully if you want to proceed with the process.

Reviewer 1 ·

Basic reporting

ok

Experimental design

ok

Validity of the findings

ok

Additional comments

ok

Reviewer 2 ·

Basic reporting

As previously written, the article describes a technical solution for a common problem in smart farms, although it uses and describes commercial hardware modules and free software. Thus, there is no new technique, hypothesis or study.

Even though the author affirms it is a development of a scientific tool, there are several other similar tools on the market, with same low cost and different technologies. The paper do not show how significant are these changes regards the other products in the market.

The author did cite PeerJ Criteria, but I would also like to remember "PeerJ Computer Science judges articles on scientific validity and suitability to join the scholarly literature".

The author did some minor improvements but did not address the main problem in the paper as it is a scientific paper.

Experimental design

As stated before, there is no relevant and meaningful scientific contribution. It is a technical solution for a problem. The author did not made the suggested modifications.

Validity of the findings

There is no finding in the paper, although it is an excellent description of a technical
solution with free software and accessible hardware.

There are several ways to deal with a technical solution with a scientific approach.

After a quick search, there are several examples below of open solutions. It also shows that this solution could have been compared with several others and it hasn't. In addition, most of these solutions have more than 5 years, are common and thus, there is no scientific validity.

"Open source hardware to monitor environmental parameters in precision agriculture" - https://doi.org/10.1016/j.biosystemseng.2015.07.005

"Irrig‐OH: An Open‐Hardware Device for Soil Water Potential Monitoring and Irrigation Management" - https://doi.org/10.1002/ird.1989

"Open Hardware: A Role to Play in Wireless Sensor Networks?" - https://doi.org/10.3390/s150306818

"SEnviro: A Sensorized Platform Proposal Using Open Hardware and Open Standards" - https://doi.org/10.3390/s150305555

Additional comments

If there is no interest in adapting to a review paper, the contribution could be handled in several other ways. But the author did not address this situation accordingly.

·

Basic reporting

No comment

Experimental design

No comment

Validity of the findings

No comment

Additional comments

I have read through the author’s responses to the reviewers and the revised version of the manuscript. I feel that my original comments have been addressed appropriately, and that the author’s responses to the other reviewers’ comments were appropriate. Overall the manuscript has been improved, and should be a useful contribution to the literature.

---

## Author Rebuttal · Round 0.2

Prof. Dr. rer. nat. Robert Winkler, CINVESTAV Unidad Irapuato
Km. 9.6 Libramiento Norte Carr. Irapuato-León, 36824 Irapuato Gto., México
Robert.Winkler@cinvestav.mx, Tel.: +52-462-6239-635

## PeerJ Computer Science

Reply to comments

MeteoMex: Open infrastructure for networked environmental monitoring and agriculture 4.0

Irapuato, October 2020

Dear Prof. Reyes-Menendez and dear Peer Reviewers,

Thank you very much for your feedback which helped to improve my manuscript.

Following, I provide a point-by-point reply to all comments:

# Editor comments (Ana Reyes-Menendez)

This work is interesting and novel but requires improvements to justify publication. Address each reviewer comment carefully, particularly those comments from Reviewer 2.

[# PeerJ Staff Note: It is PeerJ policy that additional references suggested during the peer-review process should only be included if the authors are in agreement that they are relevant and useful #]

[# PeerJ Staff Note: Please ensure that all review comments are addressed in a rebuttal letter and any edits or clarifications mentioned in the letter are also inserted into the revised manuscript where appropriate. It is a common mistake to address reviewer questions in the rebuttal letter but not in the revised manuscript. If a reviewer raised a question then your readers will probably have the same question so you should ensure that the manuscript can stand alone without the rebuttal letter. Directions on how to prepare a rebuttal letter can be found at: https://peerj.com/benefits/academic-rebuttal-letters/ #]

**RW**: I answered all reviewers' comments. With respect to reviewer 2, I would like to remind the **Editorial Criteria of *PeerJ***:

*"Decisions are not made based on any subjective determination of impact, degree of advance, novelty, being of interest to only a niche audience, etc."*
https://peerj.com/about/editorial-criteria/

The three reviewers agree about the technical quality of the manuscript, and the peers only recommend changes of the text. Thus, the paper should be acceptable in the revised version.

# Reviewer 1 (Anonymous)

## Basic reporting

The article should include a section named Background to demonstrate how the work fits into the broader field of knowledge.
Relevant prior literature must be referenced, for example:
Mulla, D. J. (2013). Twenty five years of remote sensing in precision agriculture: Key advances and remaining knowledge gaps. Biosystems engineering, 114(4), 358-371.
Saura, J. R., Reyes-Menendez, A., & Palos-Sanchez, P. (2019). Mapping multispectral Digital Images using a Cloud Computing software: applications from UAV images. Heliyon, 5(2), e01277.
Seelan, S. K., Laguette, S., Casady, G. M., & Seielstad, G. A. (2003). Remote sensing applications for precision agriculture: A learning community approach. Remote sensing of environment, 88(1-2), 157-169.

**RW**: Dear Reviewer 1, thank you very much for your work and for your suggestions. The introduction is already quite comprehensive, thus I would not add an additional Background section. However, I integrated the additional references into the introduction.

## Experimental design

The research work is clearly defined. It is relevant and meaningful.

**RW**: I am glad you liked the paper.

## Validity of the findings

The investigation was conducted rigorously but the author should improve the conclusion.

**RW**: I made the scope of the project more clear at the final of the introduction section. Now, the conclusions fit better to the initial idea of MeteoMex: An open system with professional-grade performance.

# Reviewer 2 (Anonymous)

## Basic reporting

1) The paper has several strong points, as it has a good literature review and is very didactic about the proposed system implementation. However, there is one major problem as there is no scientific contribution.

The article describes a technical solution for a common problem in smart farms, although it uses and describes commercial hardware modules and free software. Thus, there is no new technique, hypothesis or study.

This is notable in the text, as there is no problem stated, no highlights of the paper scientific contribution and its main objective.

**RW**: Dear Reviewer 1, in computational science and instrumental analysis, the development of new research tools is a crucial part. This project started with the need for a reproducible and scalable IoT platform for research projects, e.g. the monitoring of environmental conditions in field conditions in Mexico (i.e. arid areas with extreme temperatures). Different IoT systems are commercially available, but they use proprietary standards and are expensive. Community projects, on the other side, are not easily scalable, or do not comply with current standards (such as data transfer through existing WiFi networks).
The MeteoMex platform is a technical solution, which complies with professional needs in production and research, but using an open, modular and cost-efficient approach. The sensor units can be easily adapted to the users or researcher's needs and integrated in the IoT informatics platform.
Instead of answering an hypothesis, the MeteoMex provides the research tool necessary for various projects in environmental and agricultural science.
I added a paragraph at the end of the introduction, to explain better the intentions of the project.

2) The introduction does provide sufficient background and include several relevant references. It is one of the strongest points of the paper. Most of references are from the past 5 years and there is an excellent review of them. However, I would recommend avoiding some parts as L49-53 (perishable food) and L60-64 (factories examples), they are a bit off the paper main topic. An explanation of smart farms and its context would be great before L65. In addition, the introduction seems to be missing an ending paragraph to conclude the idea and relate to the paper's main topic.

**RW**: I added a paragraph at the end of the introduction, which summarizes the main goals of the project. Food storage and energy optimization are important issues that could use the same IoT technology for reducing the waste of food, water and energy with little effort. Therefore, I suggest to leave these parts. In the revised manuscript, I provide definitions of *smart farming* and *agriculture 4.0* in the introduction.

3) English language and style are fine, with just a few minor spell checks required. However, some paragraphs are fragmented, thus the text needs a small revision (For example, L46-47).

**RW**: Thanks, I corrected the sentence in L46-47 and various other writing issues.

4) Most of figures are good with enough information. One exception is the text in figures 6, 7, 8 and 11, which is not legible in the pdf.

**RW**: I agree. Those screenshots of the ThingsBoard dashboards are difficult to read. This is the result of capturing the screens and compressing the PDF. For the production of the article PDF and HTML versions I will provide the best possible resolution of the graphics. From the MeteoMex webpage (http://www.meteomex.com), the live dashboards of the examples can be accessed.

## Experimental design

It is an interesting topic and matches the paper scope. Although it isn't an easy implementation and requires a lot of study and tests, there is no relevant and meaningful scientific contribution. It is a technical solution for a problem.

A solution would be to make a comparison study with other solutions and actually test its performance or propose a new structure in communication. There are several solutions for this application with better technologies, using Zigbee and Lora networks (for example). However, it would still be hard to find a scientific gap for the study.

My suggestion for the author is to change the paper scope/type to a literature review. The paper review is excellent and updated. There aren't many reviews about these systems lately and would be great with your example of implementation with MeteoMex.

For this, only a few things have to change. For example, an expansion of the literature review as a "first part", an example of sensor module construction with MeteoMex for a second part and some "study cases" as a third part.

**RW**: Please refer to my answer above, which also makes the knowledge gap more clear (Basic reporting, 1)). The aim is not to provide another IoT protocol, but a simple, robust and scalable platform, which adopts to current standards. My paper should document and explain a novel research tool.

## Validity of the findings

There is no finding in the paper, although it is an excellent description of a technical solution with free software and accessible hardware.

**RW**: Yes, the paper describes the building of an open and scalable IoT platform with application examples.

## Comments for the Author

I believe the paper is not suitable for publication as it is now. However, I suggest to turn it to a literature review paper as I described in 2.

**RW**: As mentioned, the aim of this paper is not a literature review, but providing an open, but scalable and professional IoT platform to the community.

# Reviewer 3 (Luke Miller)

## Basic reporting
No comment

## Experimental design
No comment

## Validity of the findings
No comment

## Comments for the Author
The MeteoMex project appears to be a fairly complete set of design files and software examples necessary for making a functional sensor reporting device. I have raised some issues in the specific comments below that I feel would make the manuscript more readable and useful to the interested reader. While the files are made available on Github (a commercial site with an uncertain long-term existence), I would recommend that you also create a permanent archived copy of the existing repository on a long-term archiving site such as zenodo.org, which provides free archiving. It's certainly appropriate to provide updated files on Github on an ongoing basis.

**RW**: Dear Prof. Miller, thank you very much for your detailed review and constructive comments. Following your suggestion, I deposited the code at Zenodo:

Robert Winkler. (2020, October 9). robert-winkler/MeteoMex: PeerJ Computer Science Release v1.1 (Version v1.1). Zenodo. http://doi.org/10.5281/zenodo.4075278

I also included this information in the revised manuscript (Code availability in the methods section and References).

**LM**: The introduction could be more concise, as the listing of the various challenges of food production, storage, wastewater treatment, etc. seem to go on for a long time before the crux of the project is reached, which is a system to help monitor various aspects of these various industries and environments.

**RW**: Well, the introduction is certainly dense, but I wanted to point out the importance of monitoring environmental parameters. Very few sensor types can cover a huge variety of applications. Nevertheless, for the fast readers, I added a paragraph at the end of the introduction that comes to the point about the aim of the project.

**LM**: Line 19 Abstract: "analog, digital and I2C sensors" should be reworded, since in this context I2C is a digital protocol and a sensor using the I2C protocol is not different from the generic term "digital sensor".

**RW**: This is true. I changed the text to "analog and digital sensors."

**LM**: Line 46-47: This is a sentence fragment

**RW**: Thanks. I corrected the sentence.

**LM**: Line 57-58: This sentence doesn't seem relevant to the thrust of the paper. It could be removed.

**RW**: I think, the sentence is important. The global datasets help to predict climate and large tendencies. However, the MeteoMex project provides local data, which are more accurate for specific environments (e.g. a cow stable or greenhouse).

**LM**: Line 73: agriculture 4.0 is used here without a useful definition. The only definition I find appears at the end of the paper at line 380.

**RW**: In the revised manuscript, I provide definitions of *smart farming* and *agriculture 4.0* in the introduction.

**LM**: Line 86: AA battery does not need periods after each A.

**RW**: Thanks, corrected.

**LM**: Line 88-90: It's not immediately clear what the purpose or service provided by Blynk, ThingSpeak, MQTT is, for a reader not already familiar with those services. This sentence should be split into two, and the role of these services should be more clearly explained. The MQTT acronym should be defined.

**RW**: I added the information that Blynk and ThingSpeak are IoT platforms and defined MQTT (Message Queuing Telemetry Transport).

**LM**: Line 90-91: This sentence conveys no useful information and could be cut to make the introduction more concise.

**RW**: True, but I want to point out that already huge variety of projects exists and that I am aware of them.

**LM**: Line 93-94: This sentence gives a brief description of the attributes of the MeteoMex project, but no description of the specific capabilities (i.e. what can/could it measure, how frequently, what kind of power requirements). Those capabilities are described in more detail in section 2, but a brief rundown should be provided in the introduction.

**RW**: Thanks! I re-wrote the last paragraph of the introduction, which now clearly states the aims of the MeteoMex project.

**LM**: Figure 1: Which part the term "shield" is referring to in the caption isn't self-evident until the reader has read the text and figured out that the PCB referred to in the caption is the shield. I would suggest changing the initial sentence of the caption to read "A) circuit and B) PCB "shield"."

**RW**: Thank. I added "shield" to the caption.

**LM**: Line 99: The units of the board dimensions are given as mm squared, but the number provided are the linear dimensions (mm), so the mm2 label should be changed to mm.

**RW**: OK. I changed the units to 34.2 mm x 25.6 mm.

**LM**: Line 120-121: The DeepSleep mode is mentioned a few times in the manuscript, but it is never properly explained what this mode does. The sentence on Line 121 confuses me, since the way it is written makes it sound like the jumper needs to be closed manually in order to wake the device, and that this would need to be closed by the user each time they wanted to wake the device. But normally I would assume that a deep sleep mode would have some facility to automatically

wake itself, either on a regular time interval or because of some external sensor interrupt being triggered, rather than the user needed to physically close a jumper. If the deep sleep mode works normally, this sentence should be reworded to say something like "For enabling the DeepSleep mode, the jumper J1 must be closed."

RW: Thanks. I added information about the DeepSleep mode and Jumper J1. J1 can be permanently closed by soldering in a wire bridge. However, for programming, the shield needs to be removed then (what I would recommend anyway).

LM: Line 124: The manufacturer (Maxim) for the DS18B20 should be listed.

RW: Thanks, I added the manufacturer.

LM: Line 127: From what I read, the maximum input voltage for the analog input on the Wemos is 3.2V, so it might be worth mentioning that analog sensors that use a 5V supply must still limit their output range from 0 to 3.2V to avoid damaging the analog channel on the microcontroller.

RW: This is correct. Actually, the ESP8266 is limited to 1V, but the Wemos D1 mini board features a 3.2:1 voltage divider. I added the information to the manuscript.

LM: Line 131: The manufacturer and part number for the turbidity sensor should be provided.

RW: I added the provider information for the turbidity sensor. Such sensors are very generic (domestic dishwashers and washing maschines), and in this case  there is no part number.

LM: Line 132: The manufacturer for the JSN-SR04T sensor should be provided.

RW: I added the provider information.

LM: Line 158: change "why" to "so"

RW: Thanks. Corrected.

LM: Line 164-165: Awkward phrasing for the sentence beginning "Programming a DeepSleep/WakeUp…"

RW: Indeed. I rewrote the sentence.

LM: Line 166: What is an accumulator in this context? A battery?

RW: Yes. I changed "accumulator" to "battery".

LM: Line 239: Using a neural network to estimate solar irradiance from temperature humidity and pressure seems overly complicated and not relevant to this project. A lux or PAR sensor (though PAR sensors are expensive) could give similar information on irradiance and also interface to the MeteoMex.

RW: I completely agree and added this possibility.

LM: Lines 247-253: The process of calibrating a soil moisture sensor is glossed over here. Please provide a reference to give a more detailed description of a proper calibration routine.

RW: You are right. I added different options for calibrating the sensor.

**LM**: Line 254-256: After giving the soil temperature in Celsius, the units change to Kelvin for the subsequent temperature values. Why not continue using Celsius, especially if they are interchangeable with respect to numeric value?

**RW**: By convention, temperature differences are usually expressed using the SI unit Kelvin.

**LM**: Line 260: I assume the temperature peak was at 12 pm, not am.

**RW**: Thanks! You are completely right. I corrected this mistake.

**LM**: Line 261: The phrasing of this sentence is awkward. It would read better as "… point is only exposed to direct sunlight for a short period of the day."

**RW**: I improved the phrasing following your suggestion.

**LM**: Line 270: "economical" instead of "economic"

**RW**: Thanks. Corrected.

**LM**: Lines 272-281: I'm not sure that the example of monitoring air temperature a server room merits a separate section and figure here. Monitoring air temperature was adequately covered in section 3.3. I think most of section 3.4 could be cut to help make the paper more concise, and a mention of the server room air temperature results could be made in a sentence in the paragraph on lines 265-269 where another air temperature monitoring usage is described.

**RW**: I would like to leave the example in the manuscript, because it demonstrates the application of MeteoMex devices for monitoring the ambient conditions in industrial/ productive environments.

**LM**: Line 329: change "importing" to "important"

**RW**: Thanks. Corrected.

I hope that you are satisfied with the modifications of the manuscript and consider its publication in *PeerJ Computer Science*.

In case of any additional questions, please do not hesitate to contact me.

Yours sincerely,

Robert Winkler

---

## Round 0.3 · accepted · Accept

After reviewing your paper, following the average evaluation of the reviewers, I have decided to accept your paper for publication.

---

## Author Rebuttal · Round 0.3

Prof. Dr. rer. nat. Robert Winkler, CINVESTAV Unidad Irapuato
Km. 9.6 Libramiento Norte Carr. Irapuato-León, 36824 Irapuato Gto., México
Robert.Winkler@cinvestav.mx, Tel.: +52-462-6239-635

# Cinvestav
endorse

MeteoMex: Open infrastructure for networked environmental monitoring and agriculture 4.0

Irapuato, 25 of November 2020

Dear Prof. Reyes-Menendez and dear Peer Reviewers,

Thank you very much for reading the revised manuscript again.

Following, I reply to your comments:

# Editor comments (Ana Reyes-Menendez)

Dear author, although most of the comments from reviewer 1 and 3 were addressed, reviewer 2 still has some concerns about your improvements. Address all their comments carefully if you want to proceed with the process.

**RW**: Dear Prof. Reyes-Menendez, I answered reviewer 2; However, converting the paper into a literature review or another form of an article (she/he leaves unclear which this could be) does not seem feasible to me.

In the revised manuscript, I clearly state the knowledge gap addressed (lines 105-116). With this clarification and the endorsement of the other two reviewers, the paper should be acceptable.

# Reviewer 1 (Anonymous)

## Basic reporting
ok

## Experimental design
ok

## Validity of the findings
ok

## Comments for the author
ok

RW: Thanks again for your support and for endorsing the paper!

# Reviewer 2 (Anonymous)

## Basic reporting
As previously written, the article describes a technical solution for a common problem in smart farms, although it uses and describes commercial hardware modules and free software. Thus, there is no new technique, hypothesis or study.

Even though the author affirms it is a development of a scientific tool, there are several other similar tools on the market, with same low cost and different technologies. The paper do not show how significant are these changes regards the other products in the market.

The author did cite PeerJ Criteria, but I would also like to remember "PeerJ Computer Science judges articles on scientific validity and suitability to join the scholarly literature".

The author did some minor improvements but did not address the main problem in the paper as it is a scientific paper.

## Experimental design
As stated before, there is no relevant and meaningful scientific contribution. It is a technical solution for a problem. The author did not made the suggested modifications.

## Validity of the findings
There is no finding in the paper, although it is an excellent description of a technical solution with free software and accessible hardware.

There are several ways to deal with a technical solution with a scientific approach.

After a quick search, there are several examples below of open solutions. It also shows that this solution could have been compared with several others and it hasn't. In addition, most of these solutions have more than 5 years, are common and thus, there is no scientific validity.

"Open source hardware to monitor environmental parameters in precision agriculture" - https://doi.org/10.1016/j.biosystemseng.2015.07.005

"Irrig-OH: An Open-Hardware Device for Soil Water Potential Monitoring and Irrigation Management" - https://doi.org/10.1002/ird.1989

"Open Hardware: A Role to Play in Wireless Sensor Networks?" - https://doi.org/10.3390/s150306818

"SEnviro: A Sensorized Platform Proposal Using Open Hardware and Open Standards" - https://doi.org/10.3390/s150305555
There is no finding in the paper, although it is an excellent description of a technical solution with free software and accessible hardware.

## Comments for the Author

If there is no interest in adapting to a review paper, the contribution could be handled in several other ways. But the author did not address this situation accordingly.

**RW**: Since I want to publish a technical solution to an existing problem, converting the manuscript to a review does not make sense. I do not understand what you imply with "the contribution could be handled in several other ways."

I looked at the articles you mentioned, but none of those describes a **scalable** open IoT platform. Yes, attaching a temperature sensor to an Arduino is a ubiquitous technology. In contrast, the MeteoMex platform has the following characteristics (please check lines 105-116 of the revised manuscript :

- **Scalable**. Printed circuit boards (PCB) and standard parts allow the mass production of identical sensing units. The database server can process thousands of operations per second.
- **Flexible**. The users can connect a huge variety of commercial sensors or integrate their prototypes.
- **User-friendly**. A simple design, pre-built modules, and code examples make the platform suitable for non-experts.
- **Low cost.** Generic electronic parts, the use of existing WiFi networks, and free software reduce the installation costs. The operation is economical because of low energy consumption and the possibility of self-hosting the IoT server.
- **Open**. All relevant hardware information and the software are completely documented and freely available.

I have not found such a system in the public literature, why I believe that my contribution is original.

# Reviewer 3 (Luke Miller)

## Basic reporting
No comment

## Experimental design
No comment

## Validity of the findings
No comment

## Comments for the Author
I have read through the author's responses to the reviewers and the revised version of the manuscript. I feel that my original comments have been addressed appropriately, and that the author's responses to the other reviewers' comments were appropriate. Overall the manuscript has been improved, and should be a useful contribution to the literature.

**RW**:  Dear Prof. Miller, I much appreciate your detailed review and feedback!

In case of any additional questions, please do not hesitate to contact me.

Yours sincerely,

Prof. Robert Winkler